# Hierarchical Porous MIL-101(Cr) Solid Acid-Catalyzed Production of Value-Added Acetals from Biomass-Derived Furfural

**DOI:** 10.3390/polym13203498

**Published:** 2021-10-12

**Authors:** Shengqi Liu, Ye Meng, Hu Li, Song Yang

**Affiliations:** State Key Laboratory Breeding Base of Green Pesticide and Agricultural Bioengineering, Key Laboratory of Green Pesticide &Agricultural Bioengineering, Ministry of Education, State-Local Joint Laboratory for Comprehensive Utilization of Biomass, Center for Research and Development of Fine Chemicals, Guizhou University, Guiyang 550025, China; gs.liusq20@gzu.edu.cn (S.L.); m18334267276@163.com (Y.M.)

**Keywords:** biomass conversion, MIL-101(Cr), furfural, green chemistry, heterogeneous catalysis

## Abstract

Considering economic and environmental impacts, catalytic biomass conversion to valuable compounds has attracted more and more attention. Of particular interest is furfural, a versatile biorefinery platform molecule used as a feedstock for the production of fuels and fine chemicals. In this study, the Cr-based metal-organic frameworks (MOFs) MIL-101 were modified by chlorosulfonic acid, and MIL-101 was changed into a hierarchical MOF structure with smaller particles and lower particle crystallinity by CTAB, which significantly improved the acidic sites of the MOFs. The original and modified MIL-101(Cr) catalysts were characterized by XRD, N_2_ adsorption-desorption, SEM, TEM, and FT-IR. The effects of different catalysts, reaction temperature, catalyst amount, and alcohol type on the reaction were studied. Under the action of the MOFs catalyst, a new mild route for the condensation of furfural with various alkyl alcohols to the biofuel molecules (acetals) was proposed. The conversion route includes the conversion of furfural up to 91% yield of acetal could be obtained within 1 h solvent-free and in room-temperature reaction conditions. The sulfonic acid-functionalized MIL-101(Cr) is easy to recover and reuse, and can still maintain good catalytic activity after ten runs.

## 1. Introduction

At present, fossil carbon resources are the origin of more than 95% of the world’s chemicals and materials. The use of renewable carbon resources to produce current and new generations of chemicals and materials has been highly valued [1]. The development of alternative technologies based on sustainable resources to offset the increase in fossil energy prices caused by depletion and global warming and minimize people’s dependence on fossil fuel reserves is imminent [2,3]. Obtaining fuel alternatives from renewable biomass resources is a viable route. Lignocellulose is the second most abundant component in biomass. In recent years, some monomers in lignocellulosic biomass have been widely studied for their use in deriving more useful platform molecules, such as 5-hydroxymethylfurfural (HMF) and furfural (FUR) [4,5]. These chemical compounds are prepared by the dehydration of carbohydrates and are latently acceptable candidates for many fine chemicals currently derived from petroleum resources. FUR is a proverbial renewable platform molecule obtainable from biomass [6,7]. FUR and its derivatives have many commercial applications, including solvents, polymer monomers, and chemical raw materials for agrochemicals and medicines.

To enable the fuel properties of FUR through mixing with hydrocarbon fuels, it is usually necessary to reduce the polarity and volatility of its aldehyde group through functionalization procedures such as hydrogenation and alkylation [8]. Notably, catalytic acetalization of FUR and methanol can obtain furfural dimethyl acetal (FDMA) with high oxidation degrees as well as a high octane number, calorific value, and oxidation resistance [9,10]. This type of acetal has been used as additives for ethanol fuels to lower the auto-ignition temperature, and are considered biofuels or fuel precursors with huge potential demand [11]. FDMA also has industrial applications in the pharmaceutical, polymer, and perfume industries. At the same time, an acetal reaction is also an effective way to protect the carbonyl function of organic compounds [12,13,14]. Traditionally, acetal reactions are homogeneously catalyzed by a strong acid such as H_2_SO_4_ or HCl. These mineral acids exhibit excellent catalytic effects in acetalization reactions, but result in many difficulties related to catalyst recovery, product separation, and equipment corrosion. In this sense, the development of an environmentally friendly solid acid catalyst with high activity is one of the major difficulties to be overcome [15]. Although solid acid catalysts such as phosphotungstic acid [16], —SO_3_H-functionalized silica [17], lanthanum nitrate hexahydrate [18,19], graphene oxide [20], Fe-MIL-101 [21], GaCl_3_ immobilized on imidazolium-styrene copolymers [22], Zeolite [23], and phosphorylated polyacrylonitrile fibers [24] are not harmful to the environment, some supported solid acid catalysts require strenuous synthesis procedures and cause leaching or deactivation problems. It is therefore highly desirable to manufacture solid acid catalysts of high efficiency, e.g., by modifying the pore surface with strong acid sites [25]. Although many different kinds of functional groups can be introduced into metal-organic frameworks (MOFs) [26], it is hard to have strong acidity (e.g., sulfonic acid) after immobilizing onto the materials. If the part with Brønsted acid is used to construct the MOF skeleton, deprotonation will occur in the synthetic solution, which leads to the formation of MOF skeletons composed of conjugated Brønsted base sites [27,28]. A strong acid solution must be used as a synthesis solvent to avert the deprotonation of the acid group. Despite that most of the reported multiple MOFs or PCPs are decomposed in a strong acid solution, several robust MOFs are still stable enough. MIL-101(Cr) consisting of chromium oxide clusters and terephthalic acids are highly stable even in strong acid aqueous solutions because it is synthesized in a hydrofluoric acid medium [25]. In addition, MIL-101(Cr) material is characterized by high specific surface and easy exposure of acid sites [29].

In this work, a hierarchical porous solid acid catalyst MIL-101(Cr)-SO_3_H, prepared by a one-pot modified hydrothermal method, was investigated for the catalytic acetalization of FUR. MIL-101(Cr)-SO_3_H bears good chemical and hydrothermal stability, and has a high surface area and excellent porosity properties, with unique 3D periodic mesocrystalline cages for the satisfactory decentralization and feasible accessibility of the —SO_3_H acid site to exhibit high performance in the catalytic acetalization of FUR under solvent-free conditions. Pristine and modified MIL-101(Cr) catalysts were characterized by XRD, SEM, TEM, FT-IR, and N_2_ adsorption, and the influence of the main reaction parameters was examined. Particular concern was paid to evaluate the reusability of MIL-101(Cr)-SO_3_H catalysts. In addition, the generality for synthesizing various furfural alkyl acetals by reaction with different alcohols was established.

## 2. Materials and Methods

### 2.1. Materials

Analytical grade terephthalic acid (99%), HF (49% in water), cetyltrimethylammonium bromide (CTAB, >99%), naphthalene (99%, internal standard), furfural (99%), chromium nitrate nonahydrate (Cr(NO_3_)_3_·9 H_2_O, 99%), and ClSO_3_H were obtained from Beijing Chemical Reagent Company. Amberlyst-15 was purchased from Shanghai Aladdin Industrial Co., Ltd. (Lanzhou, China); Dichloromethane, methanol, ethanol, *n*-propanol, *n*-butanol, and isopropanol (AR) were purchased from Zhiyuan Chemical Reagent Company, Tianjin, China. Deionized water was prepared by Milli-Q Advantage A10 (Burlington, MA, USA) instrument. All reagents were purchased and used directly without any pretreatment. 

### 2.2. Preparation of Catalyst

#### 2.2.1. Preparation of Hierarchical Porous MIL-101(Cr)

According to reported methods [30,31] with slight modification, hierarchical porous MIL-101(Cr) was prepared by hydrothermal treatment of H_2_BDC (4 mmol, 0.66 g), Cr (NO_3_)_3_ 9H_2_O (4 mmol, 1.6 g), the supramolecular templating agent CTAB (1.2 mmol, 0.44 g), and HF (0.13 mL), in deionized water (20 mL). After stirring at room temperature for 30 min, the obtained mixture was put into a 50 mL stainless steel autoclave lined with a Teflon, heated to 220 °C, and held for 8 h. After the reaction, the autoclave was cooled to room temperature, and the reaction product was extracted three times with ethanol and deionized water to remove impurities and unreacted materials from the framework. Then, the purified MIL-101(Cr) (**MC**) was dried at 150 °C overnight, and stored in a desiccator after cooling to room temperature. Under the same conditions, the catalyst prepared without CTAB is defined as **MA**.

#### 2.2.2. Sulfonation of Hierarchical MIL-101(Cr)

The MIL-101(Cr) (0.33 g) was activated at room temperature by stirring in 10 mL CH_2_Cl_2_ for 20 min. To the resulting mixture, a CH_2_Cl_2_ solution containing ClSO_3_H (0.1 mL ClSO_3_H per 0.16 g solid) was slowly added dropwise under stirring conditions. After 4 h, the obtained solid was washed and filtered with ethanol and CH_2_Cl_2_ until the lotion was neutral, followed by drying to obtain mesoporous MIL-101(Cr)-SO_3_H (**MCS**) [15,32]. **MA** was functionalized with sulfonic acid under the same conditions to obtain **MAS**.

### 2.3. Characterization of the Catalyst

Powder X-ray diffraction (XRD) analysis was obtained with a Bruker D8 Advanced X-ray diffractometer. The samples’ chemical composition, morphology, and elemental diagrams were analyzed by field emission scanning electron microscopy (FE-SEM) (Zeiss SUPRA 55) at 20 kV. The outcome of the transmission electron microscopy (TEM) was furnished by a combined Philips Tecnai 20 and a JEOL JEM-2010 HR-TEM at 200 kV. First, we performed a degassing of the sample at 150 °C for five h, and then determined the nitrogen adsorption-desorption isotherm of the model at −196 °C through the Micromeritics ASAP 2020 system. The information obtained from the isotherm was used to determine the micropore volume, specific surface area, average micropore size, total pore volume, and pore size distribution. The extent of sulfonation and sulfonic acid amounts of **MCS** were detected by acid−base titration using a saturated NaCl solution to confirm the ion exchange. **MCS** (0.5 g) was suspended in 20 g of aqueous NaCl saturated solution. The mixed suspension was stirred at room temperature for more than 24 h to reach equilibrium, using 30 mL of deionized water for the next filtration and washing. Eventually, the achieved filtrate was titrated by a 0.1 M NaOH solution. The Fourier transform infrared spectrum (FT-IR) of the sample was recorded by a Nicolet 360 infrared spectrometer at 4000–500 cm^−1^, and KBr pellets were used to prepare the sample.

### 2.4. General Procedure of Catalytic Reaction and Product Analysis

All catalytic experiments were carried out at room temperature (25 °C) under vigorous stirring in a 15 mL Ace pressure tube. In a typical step, 96 mg of furfural (1 mmol), 20 mg of **MCS**, 16 mg of naphthalene (0.125 mmol, internal standard), and 3 mL of methanol were loaded into a 15 mL reaction tube. The reaction solution was stirred violently at 25 °C for 1 h. The solid catalyst and supernatant were separated by centrifugation. The reaction mixture was analyzed afterward. The supernatant after centrifugation was filtered through a membrane to remove the remaining catalyst, and then the samples were analyzed qualitatively and quantitatively using gas chromatography-mass spectrometry (Agilent 6890-5973) and Agilent 7890 B gas chromatography. The conversion was obtained by comparing the GC peak areas of products and substrates with the calibration curves of standard samples.

To study the recyclability of **MCS**. The catalyst was recovered at the end of the reaction, the identical experimental conditions as aforesaid were used. The reaction duration at every turn was 60 min, and the reaction was essentially complete. The catalyst was washed three times separately with 30 mL of methanol, then dried in a vacuum oven at 80 °C for 12 h and directly reused in the next cycle.

## 3. Results and Discussion

### 3.1. Characterization of Catalysts

To reveal the crystal structure of the synthesized catalysts, the XRD technique was carried out. Figure 1 shows the XRD patterns of **MA** prepared without CTAB, **MC** synthesized in the presence of a CTAB/Cr^3+^ molar ratio of 3, and the materials MS and **MCS** after sulfonation. The XRD pattern of the prepared **MA** was in high coincidence with the simulated XRD pattern, and all the diffraction peaks of **MAS**, **MC**, and **MCS** were in accordance with the standard literature values [33], indicating that the MOF material was successfully prepared. Furthermore, it can be observed from the figure that the XRD pattern of **MAS** was similar to that of the original **MA**, showing that the crystal structure of **MAS** did not undergo any significant changes after the insertion of the —SO_3_H group into the **MA** backbone. 

The diffraction peaks become weaker and broader in Figure 1d,e, indicating smaller size particles and lower particle crystallinity in the sample. With the addition of CTAB, the crystallinity decreased and the structure became more irregular, which is consistent with previous reports [34].

The pristine and modified MIL-101(Cr) structures were characterized, and the SEM images are shown in Figure 2a–d. FE-SEM images of **MA** without CTAB addition demonstrate a typical octahedral morphology with nanocrystals of approximately 1 μm in size (Figure 2a) [31], which correspond to pure MIL-101(Cr). By adding a certain amount of CTAB, samples with hierarchical porous structures different from the initial morphology were obtained. In addition, regular clusters of crystals about 600 nm in length and 100 nm in width were observed (Figure 2c), indicating that the mesoporous walls were formed by crystalline microporous frameworks. The observations show that CTAB has a very significant effect on MIL-101(Cr) crystal morphology, resulting in the outer surface of the nanocrystals becoming rough, which is consistent with the XRD observations. However, the sulfonic acid functionalization did not similarly affect the morphology (Figure 2b,d). Still, a significant etching of the sample can be seen, with a slight decrease in the size of the crystal particles [33].

Figure 3 shows TEM images of the structure of hierarchical porous MOFs. The uniform bright and dark spots at 20 nm in Figure 3a, c show the presence of nanocrystals guided by cetyltrimethylammonium bromide. The appearance of bundled crystals indicates that the mesopore structure is basically consistent with the SEM images. The appearance of bunched crystals indicates that the mesopore wall is composed of uniform nanocrystals and random connections without long-range and regular order, which is consistent with the disordered wormhole structure reported in SEM images and in the literature [35]. Figure 3b,d show that the catalyst appears in bundles and partially exists in the form of particles. It is further speculated that the action of CTAB causes most of the catalysts to reorganize in bundles and form an ordered hierarchical porous material, with the remaining part existing as microporous particles.

The diameter distribution of **MCS** was estimated using the software Nano measure 1.2, based on 60 points in the TEM images, and the average particle size was found to be about 41.1 nm (Figure 4). To explore the role of CTAB in the preparation of catalysts, the particle size distribution of the **MA** and **MC** catalysts was estimated based on 60 points in the TEM images. The results show that CTAB can reduce the particle size of materials (Figure 5).

The FT-IR spectrum of **MC** indicated that the modification of **MCS** was successful, as shown in Figure 6a. All peaks of the original **MC** appear in the FT-IR spectrum of the modified **MCS**. The results also indicate that the modified **MCS** can maintain the basic structure. The characteristic peak at nearly 570 cm^−1^ is associated with Cr—O stretching vibration. Most of the bands between 600~1600 cm^−1^ are attributed to H_2_BDC and its aromatic ring. Peaks at 750, 884, 1017, and 1160 cm^−1^ belong to the vibrations of C—H groups in —CH_3_, and peaks at 1508 cm^−1^ belong to C=C stretching vibrations. Peaks at 1173, 1069, and 670 cm^−1^ are attributed to an O=S=O symmetric stretching vibration, and the peak at 1702 cm^−1^ should be attributed to the S species. Therefore, it can be easily inferred that the S species was introduced in the form of —SO_3_^−^. In the absence of characteristic peaks of benzene rings replaced by sulfonic acid or sulfhydryl groups in the FT-IR spectra, all the S species imported were substituted with the monocarboxylic acid regulators ligated to the Cr cluster. 

The porosity and structural parameters of the hierarchical porous MOFs materials were characterized using N_2_ adsorption-desorption isotherms at −196 °C (Table 1). The hierarchical porous structures of ClSO_3_H-functionalized and untreated MOF showed adjustable SBET, pore volume, pore size, and acid content after the addition of a certain amount of CTAB. As listed in Table 1, both CTAB and ClSO_3_H functionalization affected the pore size of the MOFs materials, and the pore size of these porous MOFs increased from 1.71 and 1.38 nm to 3.25 and 7.32 nm after CTAB treatment, respectively. This phenomenon was due to the accumulation of nanomaterials obtained [36,37]. After sulfonic acid functionalization, the acid content of **MAS** and **MCS** reached 0.36 and 0.65 mmol (H^+^)/g, respectively. Thus, the pore volume and Brunauer–Emmett–Taylor specific surface area (BET) of MOFs without functionalization of ClSO_3_H decreased rapidly, showing a more rapid downward trend than that of functionalized porous films. Transmission electron microscopy and pore size distribution (estimated from the Horvath–Kawazoe (HK) method) analyses showed that the catalyst structure consisted of good mesopores and micropores, which were identical with the XRD and SEM characterization.

The N_2_ adsorption-desorption isotherms of **MCS** are shown in Figure 6b. In previous reports, the P/P_0_ range of the BET assay was considered to be 0.005~0.05. However, because **MCS** has microporous and mesoporous pore structures, the P/P_0_ (P/P_0_ relative pressure) in the BET test ranged from 0.05–0.23. In general, the presence of hysteresis return lines at high relative pressures may be a sign of the dominance of structurally mediated pores. According to the IUPAC classification, the isotherms shown in Figure 6 can be classified as type IV isotherms. The pore size distribution of the catalyst (Figure 6) is identical with the hysteresis loop, which affirms the existence of mesopores in the material. All these experimental results underline the pivotal role of CTAB in the formation of micropores and good mesopores.

### 3.2. Effect of Different Catalysts on the Acetalization of FUR with Methanol

First, experiments on FUR acetalization catalyzed by different acidic catalysts were carried out in methanol. Of these catalysts, the catalytic and physical property results are shown in Table 2. It is worth noting that these solid acid catalysts were employed at the identical H^+^ cation concentration. Control tests without any catalyst showed almost no acetal production, and no progression of the reaction was observed even with large excesses of methanol or extended reaction times (Table 2, entry 1). It shows that a catalyst is required to catalyze the acetal reaction. When **MA** was used as the catalyst, the existence of Lewis acid centered on chromium atoms improved its catalytic activity, but the monocarboxylic acid ligands coordinated on chromium atoms reduced the quantity of Lewis acid centers and further decreased the catalytic property (Table 2, entry 2). When **MC** was used, there is a certain but not significant increase in the yield (Table 2, entry 3), which is attributed to the increase in pore size and the probability of molecules entering the pore channel. However, when **MCS** was used as a catalyst, the modified monocarboxylic acid ligands were exchanged by —SO_3_H ligands, and more substantial acidic sites were introduced. As a result, the FDMA yield (91%) was significantly improved (Table 2, entry 4), and the high catalytic activity of the catalyst was due to the highly decentralized acid sites and the hierarchical porous structure of **MCS**. The mesopore ensures free access of molecules to the active site.

Interestingly, despite applying the same amount of acid catalyst, the catalytic effect was not so significant. Amberlyst-15, which has a relatively high number of acid sites, also gave only 63% FDMA yield (Table 2, entry 5). On the other hand, H-USY gave a higher yield to FDMA than that offered by Amberlyst-15 (Table 2, entry 6). It can be seen that the formation of FDMA was not decided entirely by the number of acid centers present in the catalyst, but also the exposure of the acid sites. In addition, our catalyst was compared with the previous MOF material. According to the data of Cu_3_(BTC)_2_ catalyzed furfural acetal reaction reported by Dhakshinamoorthy et al. [34], the catalytic effect of our developed catalyst **MCS** was better than that of Cu_3_(BTC)_2_ (Table 2, entry 4 vs. 7). In all situations, FDMA was the unique product formed, and the gas chromatography detected the only product in the reaction. It is well known that the synthesis of FDMA needs to start from the formation of the corresponding hemiacetal. Still, no intermediates were detected in any reaction assay of this system. This indicates a faster and more complete build of acetals than hemiacetal formation, which agrees with other previously reported techniques.

### 3.3. Effect of Catalyst Dosage on the Acetalization of FUR with Methanol

Acetals are formed by acid-catalyzed reversible reactions, in which the catalyst is used to enhance the electrophilicity of FUR carbonyl carbon, thus allowing the reaction to proceed. The effect of catalyst dosage on the acetalization of FUR was investigated under typical experimental conditions (Figure 7a). By and large, the rise in the amount of catalyst means an increase in the number of acid-active sites usable for the catalytic procedure. Therefore, the conversion (i.e., reaction rate) is anticipated to rise if the reaction is under kinetic control. When a small amount of catalyst (10 mg) was applied, the yield of the FDMA arrived 48% after 1 h of reaction, with some improvement in yield after a longer reaction time. This confirms that the catalyst maintains activity and requires only a longer reaction time to produce nearly the equivalent yield of FDMA, as an enormous amount of catalyst is present in a shorter reaction time. However, when higher loading catalysts were used, the yield of FDMA decreased to 86% and 83% when 40 mg and 50 mg **MCS** catalysts were used, respectively. The decrease in FDMA yield may be due to the excess acid that can protonate methanol and significantly reduce its nucleophilicity. Also, acid-catalyzed side reactions may occur, for instance, in the resinification of FUR.

### 3.4. Effect of Temperature on the Acetalization of FUR with Methanol

The catalytic performance of **MCS** for methanol FUR acetalization at different temperatures was investigated. As shown in Figure 7b, the main observation is that although the initial rate of the reaction is affected by the rapid increase of temperature, the final yield is very close. When the reaction is carried out at a higher temperature, the number of effective collisions increases. Therefore, the products (e.g., FUR, diethyl acetal, and water) were formed rapidly. However, the reaction almost reached equilibrium after 60 min. The main reason may be that the reaction equilibrium may be transferred to the reactant in the presence of more water. This is also the main reason why the temperature had no noticeable effect on the reaction when the yield reached a particular value.

### 3.5. Substrate Test

In order to evaluate the importance of appropriate pore size in MOF structures in acetalization, as a comparative experiment, different aldehydes were catalyzed by **MCS** in methanol. The obtained results are shown in Table 3. It is reported that aliphatic aldehydes have low reactivity to the formation of corresponding dimethylacetals [38]. However, aliphatic aldehydes also showed high reactivity in the presence of the **MCS** catalyst (Table 3, entries 1–3). In addition, structurally rigid aromatic aldehydes can also be converted to corresponding dimethyl acetals with excellent conversion (Table 2, entry 6).

The length of the alcohol carbon chain and the spatial site resistance around the hydroxyl group are the primary characteristics that affect the activity of alcohol reactions. Therefore, the catalytic performance of **MCS** catalysts in the acetalization reactions of FUR with different alcohols was evaluated at room temperature. The results are summarized in Figure 8a.

In the presence of the **MCS** catalyst, the changing trend of alcohol activity was methanol > ethanol > propanol > butanol > isopropanol. The activity of furfural acetalization can be attributed to the steric hindrance of isopropanol on the hydroxyl group as the reason for the lowest reaction activity on one hand, while on the other hand, the decrease of reactivity with the increase of carbon chain length can be attributed to the *pK*a effect of alcohol. The *pK*a effect of alcohol decreases with the increase of carbon chain length, which reduces the reactivity of alcohol in FUR acetalization. Although two aspects (i.e., the spatial resistance on the hydroxyl group and the size of the carbon chain) may affect the alcohol reactivity in FUR acetalization, it is evident from the results that the spatial resistance on the hydroxyl group has a more significant effect on acetal formation than the length of the alcohol carbon. Regardless of the alcohol substrate employed, all reactions achieved complete conversion in a short reaction time, with the exclusive product being detected.

### 3.6. Catalyst Recycling Study

Reusability is a crucial characteristic of catalysts, so it is necessary to investigate the recyclability of **MCS**. Interestingly, the recovered **MCS** exhibited relatively stable catalytic activity after the initial run, with a slight decrease in activity after ten runs (Figure 8b), indicating that the ClSO_3_H-modified MIL-101(Cr) catalyst has high stability. This demonstrates that MOF is an excellent solid acid catalyst with increased stability in catalytic acetalization. The XRD patterns and FT-IR spectra (Figure 9) of the fresh **MCS** and reused **MCS** catalysts showed that the integrity of —SO_3_H ligands and crystals were well retained after ten cycles.

## 4. Conclusions

In summary, the original catalyst and the ClSO_3_H modified MIL-101(Cr) catalyst were characterized by TEM, XRD, SEM, N_2_ adsorption-desorption, and infrared spectroscopy. The results showed that the hierarchical porous catalyst **MCS** has chemical and hydrothermal stability, high specific surface area, good porosity, and a unique three-dimensional periodic cage with good dispersion and feasible accessibility of the –SO_3_H acid sites. The catalytic activity of **MCS** as a reusable solid acid catalyst for the acetalization of FUR with methanol was also studied. In the presence of **MCS**, FUR could react with methanol to form the expected dimethanol acetal with up to 91% yield in 1 h. The high catalytic activity of **MCS** may be due to the suitable Brønsted acid sites and hierarchical porous structure, which ensures that the reaction substrate can enter the MOFs freely. The **MCS** catalyst is a highly recyclable solid acid catalyst in acetalization, with potential for industrial applications because its high catalytic activity and structural integrity can be maintained after ten times of continuous use.

## Figures and Tables

**Figure 1 polymers-13-03498-f001:**
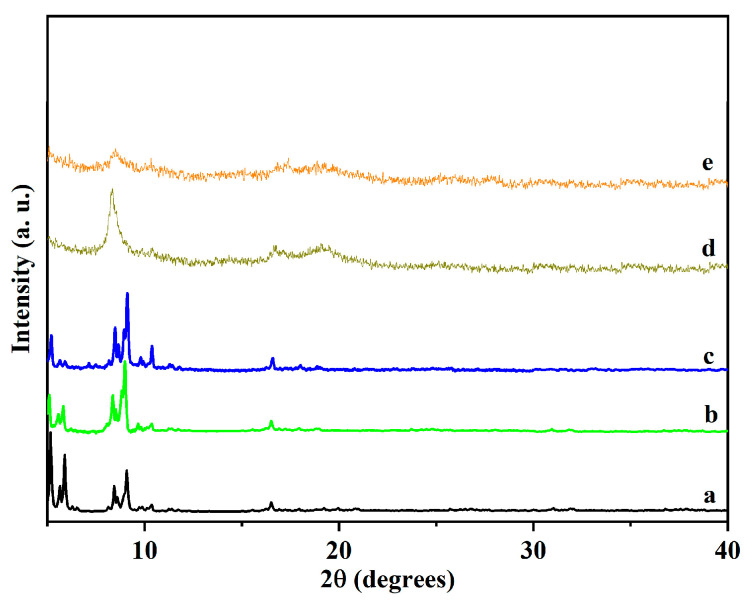
XRD patterns of (**a**) simulated MIL-101, (**b**) **MA**, (**c**) **MAS**, (**d**) **MC**, and (**e**) **MCS**.

**Figure 2 polymers-13-03498-f002:**
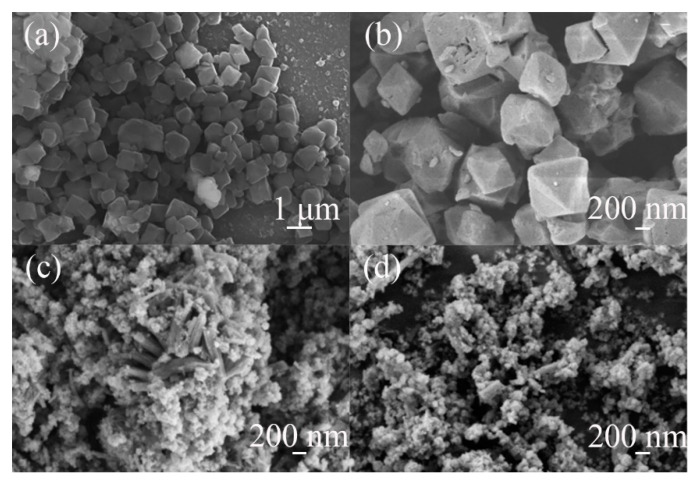
SEM images of: (**a**) **MA**, (**b**) **MAS**, (**c**) **MC**, and (**d**) **MCS**.

**Figure 3 polymers-13-03498-f003:**
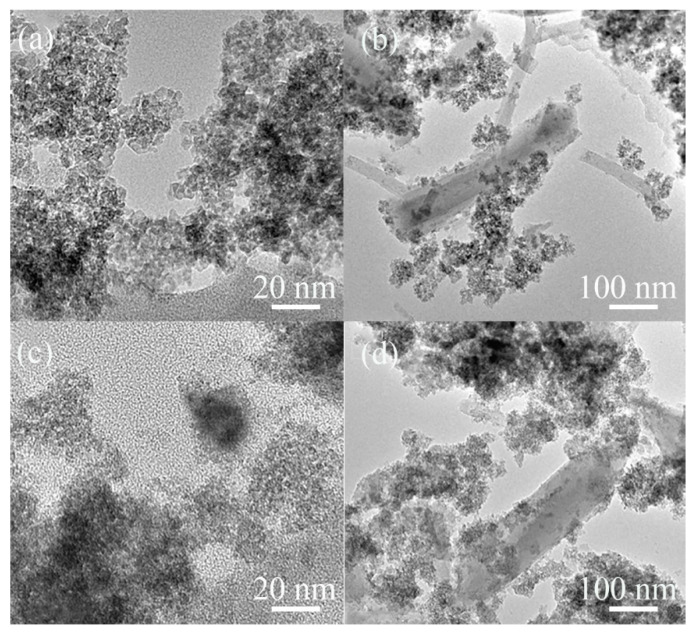
TEM images of (**a**,**b**) **MC** and (**c**,**d**) **MCS** in different scale plates.

**Figure 4 polymers-13-03498-f004:**
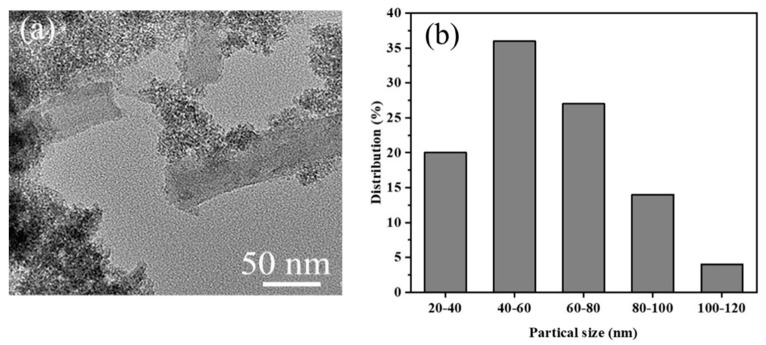
TEM image (**a**) and particle size distribution (**b**) of **MCS**.

**Figure 5 polymers-13-03498-f005:**
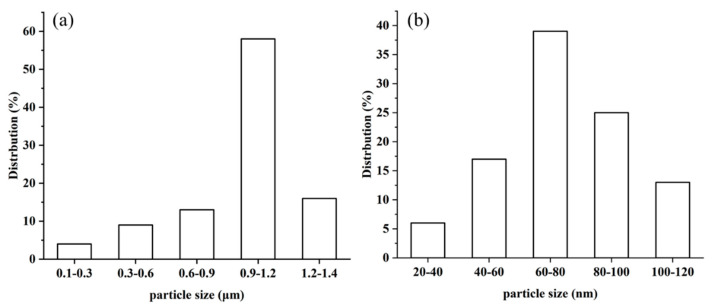
The particle size distribution of **MA** (**a**) and **MC** (**b**).

**Figure 6 polymers-13-03498-f006:**
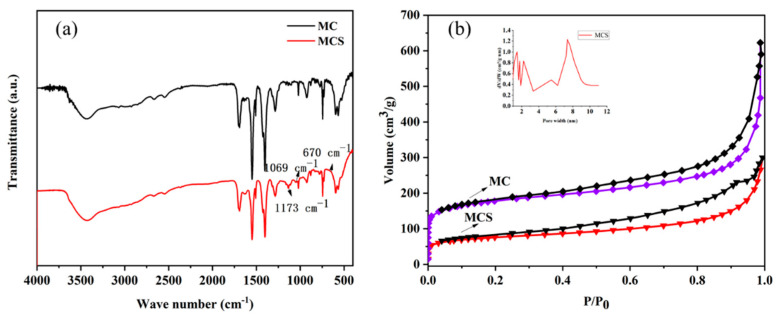
FT-IR spectra (**a**) and N_2_ adsorption-desorption isotherms (**b**) of pristine and modified MIL-101(Cr).

**Figure 7 polymers-13-03498-f007:**
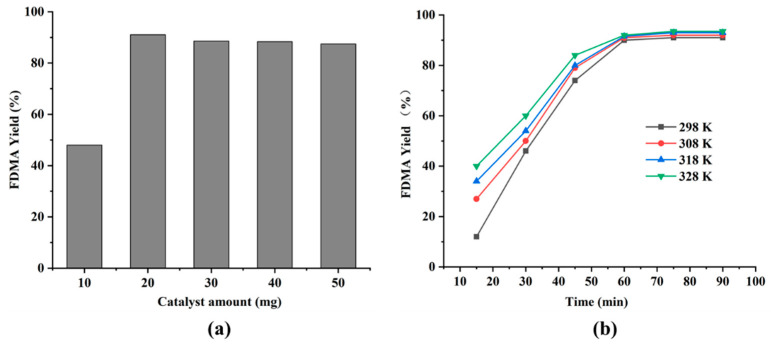
(**a**) The influence of **MCS** catalyst amount on the yield of FDMA synthesized from FUR acetalization. (**b**) The influence of temperature on the yield of the FDMA. Reaction conditions: 96 mg FUR (1 mmol), 3 mL methanol (74 mmol), room temperature for 1 h.

**Figure 8 polymers-13-03498-f008:**
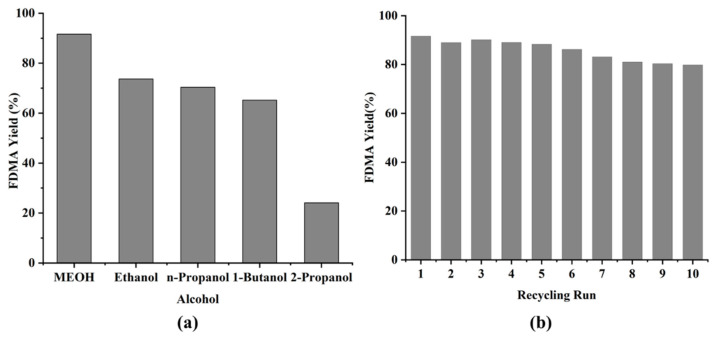
(**a**) Effect of alcohol on the FUR acetalization catalyzed by **MCS**. (**b**) Reuse of **MCS** catalyst in FUR acetalization.

**Figure 9 polymers-13-03498-f009:**
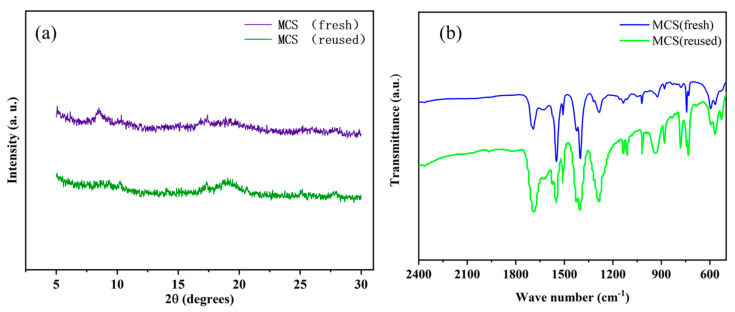
XRD patterns (**a**) and FT-IR spectra (**b**) of the fresh and reused **MCS** catalysts.

**Table 1 polymers-13-03498-t001:** Physicochemical properties of various MIL-101 catalysts.

Catalyst	Pore Size (nm)	S_BET_ (m^2^/g)	V_t_ (cm^3^/g)	V_meso_ (cm^3^/g)	V_meso_/V_micro_	A_titration_ (mmol(H^+^)/g)
**MA**	1.71	2603	1.30	0.39	0.42	—
**MAS**	1.38	1825	0.90	0.23	0.34	0.36
**MC**	3.25	638	0.51	0.28	1.22	0.03
**MCS**	7.32	197	0.30	0.18	1.50	0.65

**Table 2 polymers-13-03498-t002:**
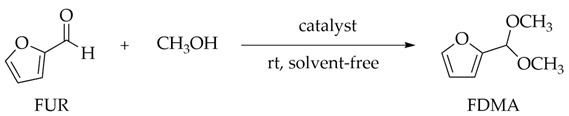
Conversion of FUR into FDMA over various catalysts ^a^.

Entry	Catalyst	S_BET_ [m^2^/g]	Pore Volume [cm^3^/g]	A_titration_ [mmol(H^+^)/g]	Catalyst Amount (mg)	FDMA Yield (%)
1	No catalyst	—	—	—	—	1
2	**MA**	2594	1.30	—	20	9
3	**MC**	638	0.51	0.03	20	25
4	**MCS**	197	0.30	0.65	20	91
5	Amberlyst-15	50	—	4.70	5	63
6	H-USY(Si/Al = 6)	37.6	0.08	1.40	5	71
7 ^b^	Cu_3_(BTC)_2_	1019	—	—	100	86

^a^ Reaction conditions: 1 mmol furfural, catalyst with 3 mol% acid content, 15 mg (0.12 mmol) naphthalene (internal standard), 3 mL methanol, room temperature, and 1 h reaction time. ^b^ Reaction conditions: FUR (1 mmol), methanol (3 mL), Cu_3_(BTC)_2_ (100 mg), 24 h, room temperature [34].

**Table 3 polymers-13-03498-t003:** Acetalization of various aldehydes was catalyzed by **MCS** with methanol ^a^.

Entry	Aldehyde	Acetal	Yield (%)
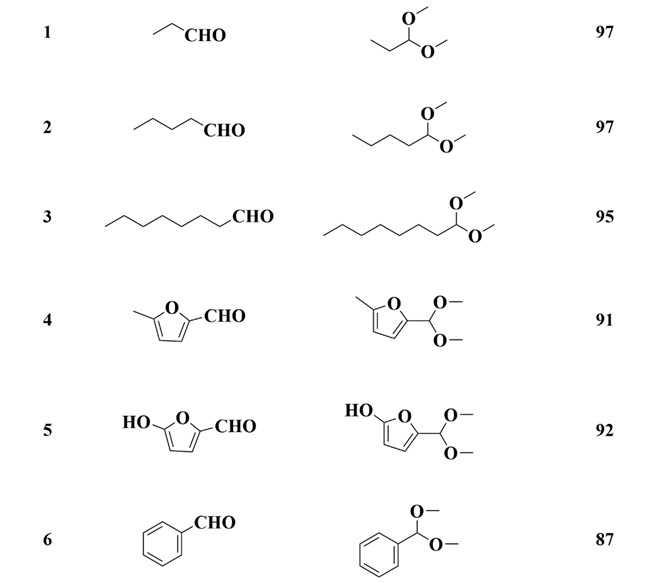			

^a^ Reaction conditions: 1 mmol aldehyde, catalyst with 3 mol% acid content, 15 mg (0.12 mmol) naphthalene (internal standard), 3 mL methanol, room temperature, and 1 h reaction time.

## Data Availability

Raw data of this work is available upon request from the corresponding author.

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
