# Peer review of "Hierarchical Porous MIL-101(Cr) Solid Acid-Catalyzed Production of Value-Added Acetals from Biomass-Derived Furfural"

_polymers, 2021, doi:10.3390/polym13203498_

Round 1

Reviewer 1 Report

This work done by Liu et al. reported the synthesis of mesoporous MIL-101 (Cr) and M-101(Cr)-SO3H and their catalytic activity for acetalization of furfural. The authors demonstrated that high yields could be obtained in the furfural acetalization when Brønsted acid sites are present in the MOF structure (M-101(Cr)-SO3H (MCS)). Additionally, the authors studied the effect of different alcohols and the amount of catalysts in the acetalization. In addition, the MCS showed high catalytic stability as demonstrated after five cycles. However, this work lacked solid proof to support the as-claimed conclusions and required several adjustments. Detailed comments or questions are listed as below:

  1. The authors claimed that “It is known from the literature that lower and smaller crystals at the nanoscale guided the formation of mesoporous MOFs materials.” Can the authors provide references that support this claim?. It is well known that nanoparticles can “guide” the formation of hierarchical MOFs materials due to the presence of micropores and mesopores. However, mesoporous MOFs materials are not the same as hierarchical MOFs materials. Indeed, MIL-101 is considered as mesoporous MOF (pore diameters close to 30 Å).

  1. In the same line, the authors claimed that “the pivotal role of CTAB in the formation of micropores and good mesopores.” However, CTAB (and other surfactants) have been used for size control and morphology of MOFs particles and, in some cases, to the synthesis of hierarchical MOFs materials (https://pubs.rsc.org/en/content/articlelanding/2019/cs/c9cs00472f). How CTAB helps the formation of micropores and “good mesopores”?. In addition, in the abstract, the authors also claim that “the dispersed MOFs micropore was expanded into an ordered mesoporous structure by CTAB. Can the micropores of the MOF be expanded to mesopores by using CTAB? I believe that the authors do not have clear the role of CTAB in MOF synthesis. The authors should clarify this point since the role of CTAB is well known in MOF synthesis.

  1. PXRD revealed that MC and MCS have very low crystallinity. This fact was attributed to the formation of nanoparticles. However, I recommend studying other ratios CTAB:Cr3+ (lower than 3). Will crystallinity improve by decreasing that ratio? How it affects the catalytic performance?

  1. Authors should include the particle size distribution of MA and MC.

  1. Authors claimed that “By adding a certain amount of CTAB, samples with mesoporous structures different from the initial morphology were obtained” MOF itself is mesoporous. This is in the same line as question 2.

  1. SEM and TEM revealed the presence of two types of crystals (Figure 2c, and Figure 3).Is there any inorganic impurity?. Can the authors provide XRD up to 2-theta = 70.

  1. The authors claimed that “crystal clusters of about 600 nm in length and 100 nm in width (Figure 2c), which state the existence of mesoporous samples and the mesoporous wall formed by crystal microporous framework”. How can the presence of crystals of around 600 nm prove the presence of mesoporosity?

  1. N2 adsorption-desorption isotherm of MC, and its BET value is significantly lower than the reported one (DOI: 10.1126/science.1116275). In fact, from the isotherm is not possible to observe the increase in the absorption in the mesoporous region characteristic of MIL-101. This confirms the poor-quality material that was obtained.

  1. The stability of MCS after reaction needs further study. Five cycles are not enough to draw a conclusion of high stability.

  1. Why the PXRD of fresh and reused MCS look more crystalline (Figure 8) than that reported in figure 1. In figure 1 looks more like amorphous material. However, figure 8 looks more like the expected MIL-101.

Thus, in the view of this reviewer, unfortunately, the impact of the present study is far below the one expected for a paper in Polymers.

Author Response

This work done by Liu et al. reported the synthesis of mesoporous MIL-101 (Cr) and M-101(Cr)-SO3H and their catalytic activity for acetalization of furfural. The authors demonstrated that high yields could be obtained in the furfural acetalization when Brønsted acid sites are present in the MOF structure (M-101(Cr)-SO3H (MCS)). Additionally, the authors studied the effect of different alcohols and the amount of catalysts in the acetalization. In addition, the MCS showed high catalytic stability as demonstrated after five cycles. However, this work lacked solid proof to support the as-claimed conclusions and required several adjustments. Detailed comments or questions are listed as below:

Response: Thank you very much for your valuable comments and suggestions. All the raised major issues have been clarified and resolved carefully, and reliable evidence has been provided to support our conclusions. We believe the revised manuscript can meet the journal's standards and requirements. It would be highly appreciated if you could kindly reconsider our manuscript for possible publication.

  1. The authors claimed that “It is known from the literature that lower and smaller crystals at the nanoscale guided the formation of mesoporous MOFs materials.” Can the authors provide references that support this claim? It is well known that nanoparticles can “guide” the formation of hierarchical MOFs materials due to the presence of micropores and mesopores. However, mesoporous MOFs materials are not the same as hierarchical MOFs materials. Indeed, MIL-101 is considered as mesoporous MOF (pore diameters close to 30 Å).

Response: Thank you very much for your kind comments. We reviewed the statement and amended it. It is undeniable that MIL-101 (Cr) is considered as mesoporous MOF (the pore size is close to 30 Å), but this pore size refers to the average pore size. It can be considered that MIL-101 (Cr) has some micropores and mesopores. At the same time, the physical properties of the materials obtained by different catalyst preparation methods are also different. Our method is similar to the preparation method reported by Mohebbi-Kalhori et al. (Chem. Eng. J. 2018, 341,164-174) and microporous MIL-101 (Cr) is also prepared (Page 4).

  1. In the same line, the authors claimed that “the pivotal role of CTAB in the formation of micropores and good mesopores.” However, CTAB (and other surfactants) have been used for size control and morphology of MOFs particles and, in some cases, to the synthesis of hierarchical MOFs materials (https://pubs.rsc.org/en/content/articlelanding/2019/cs/c9cs00472f). How CTAB helps the formation of micropores and “good mesopores”? In addition, in the abstract, the authors also claim that “the dispersed MOFs micropore was expanded into an ordered mesoporous structure by CTAB. Can the micropores of the MOF be expanded to mesopores by using CTAB? I believe that the authors do not have clear the role of CTAB in MOF synthesis. The authors should clarify this point since the role of CTAB is well known in MOF synthesis.

Response: Thank you for your comments. At present, CTAB is often used to control the size and morphology of MOFs particles. In addition, Zhang et al. (Angew. Chem. Int. Edit. 2008, 47, 9487-9491, numbered as [30]) also confirmed that CTAB can improve the pore size.

We are sorry for our negligence of relevant writing issues. In this study, surfactant micelles are believed to be responsible for the structure-oriented assembly of the mesoporous MOF. First, the deprotonated organic ligand BTC2- may enter the solvent region to balance the cationic charges on the surface of the micelles. Electrostatic interactions between cationic surfactant micelles and negatively charged BTC2- ions lead to the positioning of frame building blocks. Second, the nucleation and crystal growth processes lead to microporous MOF nanoparticles, which are self-assembled from frame building blocks (i.e., Cr2+ ions and BTC2- ligands) in a continuous solvent region between micelles. After the surfactant molecules were removed from the solid, part of the micropores were expanded into mesopores, so micropores and mesoporous MOFs were obtained.

  1. PXRD revealed that MC and MCS have very low crystallinity. This fact was attributed to the formation of nanoparticles. However, I recommend studying other ratios CTAB: Cr3+ (lower than 3). Will crystallinity improve by decreasing that ratio? How it affects the catalytic performance?

Response: Thank you for your suggestion. According to the literature (CrystEngComm, 2012, 14, 1613-1617; numbered as [33]), reducing the ratio of CTAB: Cr3+ will increase the proportion of the crystallinity. However, when the ratio of CTAB: Cr3+ is lower than 3, it basically has no effect on the MOFs material; specifically, CTAB cannot change the morphology and aperture effect.

  1. Authors should include the particle size distribution of MA and MC.

Response: Thank you for your suggestion. The particle size distributions of MA and MC are given in Figure 5 of the revised manuscript (Page 6).

  1. Authors claimed that “By adding a certain amount of CTAB, samples with mesoporous structures different from the initial morphology were obtained” MOF itself is mesoporous. This is in the same line as question 2.

Response: Thank you very much for your kind comments. It is undeniable that MIL-101 (Cr) is considered as mesoporous MOF (the pore size is close to 30 Å), but this pore size refers to the average pore size. It can be considered that MIL-101 (Cr) has some micropores and mesopores. It can be seen from the experimental data in this study that when CTAB: Cr3+=3 is added, the morphology of MOFs changes from typical octahedral shape to rough nanoparticles. The mean pore sizes range from 1.71 to 7.32 nm.

  1. SEM and TEM revealed the presence of two types of crystals (Figure 2c, and Figure 3). Is there any inorganic impurity? Can the authors provide XRD up to 2-theta = 70.

Response: Thank you for your suggestion. We have provided the XRD pattern of up to 2-theta = 70, and no impurities were found in the figure (Page 4).

  1. The authors claimed that “crystal clusters of about 600 nm in length and 100 nm in width (Figure 2c), which state the existence of mesoporous samples and the mesoporous wall formed by crystal microporous framework”. How can the presence of crystals of around 600 nm prove the presence of mesoporosity?

Response: Thank you very much for your kind comments. The presence of a crystal at around 600 nm does not prove the presence of a mesopore. It has been revised in the manuscript. The existence of mesoporous samples can only be seen through pore size distribution, and we have included the corresponding pore size distribution map (Figure 6; Pages 5, 7).

  1. N2 adsorption-desorption isotherm of MC, and its BET value is significantly lower than the reported one (DOI: 10.1126/science.1116275). In fact, from the isotherm is not possible to observe the increase in the absorption in the mesoporous region characteristic of MIL-101. This confirms the poor-quality material that was obtained.

Response: Thank you very much for your kind comments. BET is lower than the value in this paper mainly because our method makes its particle size smaller. Due to the presence of sulfonic acid in its frame, MIL-101 (Cr) is blocked by the cage, resulting in a decrease in specific surface area. We have added the mesoporous/microporous volume data of the material (Table 1). From the data (Vmeso/Vmicro (cm3/g): MA (0.42), MAS (0.34), MC (1.22), MCS (1.50)), we can see that the mesoporous increases accordingly.

  1. The stability of MCS after reaction needs further study. Five cycles are not enough to draw a conclusion of high stability.

Response: Thank you for your suggestion. We repeated it ten times and the effect decreased slightly. The corresponding data are shown in Figure 8 (Page 11).

  1. Why the PXRD of fresh and reused MCS look more crystalline (Figure 8) than that reported in figure 1. In figure 1 looks more like amorphous material. However, figure 8 looks more like the expected MIL-101.

Response: Thank you very much for your kind comments. The scale of the XRD pattern has been adjusted for clear comparison (Figure 9). MCS itself is no crystal shape, which is similar to previous reports (CrystEngComm, 2012, 14, 1613-1617; numbered as [33]). CTAB will change the overall shape of the material (Page 12).

Thus, in the view of this reviewer, unfortunately, the impact of the present study is far below the one expected for a paper in Polymers.

Response: Thank you very much again for your comments. As you kindly suggested, the manuscript has been further improved, as shown above. Many thanks.

Reviewer 2 Report

The authors have taken into account the comments from the initial submission and I have no further comments.

Author Response

Response: Thank you very much for your kind recommendation.

Reviewer 3 Report

The paper by S. Yang reports the preparation of the highly mesoporous MIL-101(Cr)-SO3H catalyst with strong Brønsted acid sites and its investigation in the producing valuable acetals from biomass-derived furfural. The studied catalytic reaction is important from practical point of view and widely studied over different solid catalysts. The paper is interesting and may contribute to the development of the efficient MOF-based catalysts for the biomass upgrading.

The paper can be published after addressing the following points.

  1. The title of this paper should be corrected. In the present form, it is rather strange.
  2. What is means “ and the dispersed MOFs micropore was expanded into an ordered mesoporous structure.
  3. Normally, MIL-101 porous structure includes two types of mesopores. It is confirmed also by its adsorption isotherm presented on the Fig. 5b. Why authors claim that is microporous? Please, present the mesopore size distribution and mesopore/micropore volume for the studied materials.
  4. According to presented XRD results (Fig. 1), the crystalline structure of the MIL-101(Cr) is destroyed after sulfonation.
  5. A couple of recent reviews on biomass upgrading using MOF-based catalysts https://doi.org/10.3390/catal8090368 and https://doi.org/10.1155/2020/1201923 may be interesting for the readers.
  6. English should be improved.

Author Response

The paper by S. Yang reports the preparation of the highly mesoporous MIL-101(Cr)-SO3H catalyst with strong Brønsted acid sites and its investigation in the producing valuable acetals from biomass-derived furfural. The studied catalytic reaction is important from practical point of view and widely studied over different solid catalysts. The paper is interesting and may contribute to the development of the efficient MOF-based catalysts for the biomass upgrading.

The paper can be published after addressing the following points.

Response: Thank you very much for your valuable comments and suggestions. All the raised major issues have been clarified and resolved carefully, which have been listed below.

  • The title of this paper should be corrected. In the present form, it is rather strange.

Response: Thank you for your suggestion. The title of the paper has been re-written in the revised manuscript “Mesoporous MIL-101 (Cr) solid acid-catalyzed production of value-added acetals from biomass-derived furfural” (Page 1). Thanks.

  • What is means “and the dispersed MOFs micropore was expanded into an ordered mesoporous structure.

Response: Thank you for your suggestion. In this study, surfactant micelles are believed to be responsible for the structure-oriented assembly of the mesoporous MOF. First, the deprotonated organic ligand BTC2- may enter the solvent region to balance the cationic charges on the surface of the micelles. Electrostatic interactions between cationic surfactant micelles and negatively charged BTC2- ions lead to the positioning of frame building blocks. Second, the nucleation and crystal growth processes lead to microporous MOF nanoparticles, which are self-assembled from frame building blocks (i.e., Cr2+ ions and BTC2- ligands) in a continuous solvent region between micelles. After the surfactant molecules were removed from the solid, part of the micropores were expanded into mesopores, so micropores and mesoporous MOFs were obtained.

  • Normally, MIL-101 porous structure includes two types of mesopores. It is confirmed also by its adsorption isotherm presented on the Fig. 5b. Why authors claim that is microporous? Please, present the mesopore size distribution and mesopore/micropore volume for the studied materials.

Response: Thank you for your suggestion. It is undeniable that MIL-101 (Cr) is considered as mesoporous MOF (the pore size is close to 30 Å), but this pore size refers to the average pore size. It can be considered that MIL-101 (Cr) has some micropores and mesopores. At the same time, the physical properties of the materials obtained by different catalyst preparation methods are also different. Our method is similar to the preparation method reported by Mohebbi-Kalhori et al. (Chem. Eng. J. 2018, 341, 164-174), and microporous MIL-101 (Cr) is also prepared. MCS pore size distribution and mesopore/micropore volume data have been given (Table 1; Page 7).

  • According to presented XRD results (Fig. 1), the crystalline structure of the MIL-101(Cr) is destroyed after sulfonation.

Response: Thank you for your suggestion. According to XRD results (Fig. 1), the peak position of sulfonated MIL-101(Cr) (MAS) is consistent with that of MA, indicating that the crystal structure is basically unaffected. However, after the addition of CTAB, the crystal structure of MC has been destroyed. Thanks.

  • A couple of recent reviews on biomass upgrading using MOF-based catalysts https://doi.org/10.3390/catal8090368 and https://doi.org/10.1155/2020/1201923 may be interesting for the readers.

Response: Thank you for your suggestion. These two articles are of great help to us, enabling us to have a more thorough understanding of the application of MOFs in the biomass upgrade process, and are quoted in the paper (Catalysts 2018, 8 ,368-407.; Adv. Polym. Tech. 2020, 2020, 1-11; numbered as [28, 29]) (Page 2).

  • English should be improved

Response: As you kindly suggested, the manuscript has been further refined, and especially we have improved the language level to make the manuscript more readable. Thank you.

Reviewer 4 Report

Manuscript ID: polymers-1344011

This work may be of interest to the readers of Polymers, however, it requires major revision before it can be considered for publication. The specific comments are

  • Authors should provide more convincing evidences for the existence of MOFs in MC and MCS (Figure 1).
  • The discussions concerning to SEM and TEM should be thoroughly rechecked and these results are not matching with XRD data. What is the purpose to use CTAB?
  • IR spectra should be reported in the range of 4000-400 cm-1.
  • Lines 236-239 should be checked carefully.
  • Exact weight for the catalysts used in Table 2 should be provided.
  • Figure 8 clearly shows that the catalyst undergoes structural change after reusing for many cycles than to the fresh solid. But, the text has to be reconsidered carefully.
  • The FT-IR and XRD of the fresh solid in Figure 8 is different from figure 5a and 1. Why?
  • Substrate scope is not tested. Authors should check various substrates with different kinetic diameter of aldehydes to probe the influence of pore size and surface area.
  • 15 and 32 are same. This has to be corrected.
  • Some recent reviews from H. Garcia on MOFs acid catalysis may be added in the introduction.

Author Response

This work may be of interest to the readers of Polymers, however, it requires major revision before it can be considered for publication. The specific comments are

Response: Thank you very much for your comments and kind recommendation. All the raised major issues have been clarified and resolved carefully, which have been listed below.

  1. Authors should provide more convincing evidences for the existence of MOFs in MC and MCS (Figure 1).

Response: Thank you very much for your kind comments. The XRD patterns of MC and MCS have been redescribed, demonstrating the presence of MOFs in MC and MCS (Page 4).

  1. The discussions concerning to SEM and TEM should be thoroughly rechecked and these results are not matching with XRD data. What is the purpose to use CTAB?

Response: Thank you very much for your kind comments. The discussion section on SEM and TEM has been carefully reviewed and revised. The results show that CTAB plays a key role in limiting the crystal size to prepare nanostructures, and CTAB can change pore size. Zhang et al. (Angew. Chem. Int. Edit. 2008, 47, 9487-9491; numbered as [30]) confirmed that CTAB can improve the pore size (Pages 5, 6).

  1. IR spectra should be reported in the range of 4000-400 cm-1.

Response: Thank you very much for your kind comments. Infrared spectra in the range of 4000-400 cm-1 are given in the revised manuscript (Figure 6; Page 7).

  1. Lines 236-239 should be checked carefully.

Response: Thank you very much for your kind comments. We have carefully checked lines 236-239 and made changes (Page 7). Thanks.

  1. Exact weight for the catalysts used in Table 2 should be provided.

Response: Thank you very much for your kind comments. The exact weight of the catalyst used has been provided (Table 2; Page 8).

  1. Figure 8 clearly shows that the catalyst undergoes structural change after reusing for many cycles than to the fresh solid. But, the text has to be reconsidered carefully.

Response: Thank you very much for your kind comments. The structure of the catalyst does change after many cycles and has been redescribed (Page 11).

  1. The FT-IR and XRD of the fresh solid in Figure 8 is different from figure 5a and 1. Why?

Response: Thank you very much for your kind comments. We are sorry for our negligence of relevant Figure issues, which was caused by analysis of different runs of catalysts. We have revised FT-IR and XRD (Page 12)

  1. Substrate scope is not tested. Authors should check various substrates with different kinetic diameter of aldehydes to probe the influence of pore size and surface area.

Response: Thank you very much for your kind comments. We added different aldehydes as substrates to explore the effect of pore size and specific surface area (Page 10).

  1. 15 and 32 are same. This has to be corrected.

Response: Thank you very much for your kind comments. Duplicate references have been deleted (Page 13).

  1. Some recent reviews from H. Garcia on MOFs acid catalysis may be added in the introduction.

Response: Thank you for your valuable suggestion. This recent reviews from H. Garcia has been cited in the introduction (Chem. Soc. Rev. 2020, 49, 3638-3687; numbered as [26]). Thanks. (Page 2)

Round 2

Reviewer 1 Report

I have carefully read the response letter and the resubmitted manuscript and, I still have some concerns about the work and do not suggest to publish it at the present form.

Indeed, my biggest concern is related with the fact that the authors claimed that they obtained mesoporous MIL-101 when the synthesis was performed in the presence of CTAB. By definition, a mesoporous MOF exhibit open channels or cavities between 2 nm up to 50 nm (at structural level). As I commented in my previous review, MIL-101 is considered itself as a mesoporous MOF since in its structure exhibits pores and cavities up to 30 A.  By claiming that using CTAB we can obtain mesoporous MOFs could be misunderstanding by the readers. However, by reading and checking carefully the characterization provide by the authors, I strongly believe that interparticle mesopores are formed due to the aggregation of the obtained nanoMOF. This phenomena is not new and it was widely observed when nanoMOFs are synthesized, especially in presence of surfactants (DOI https://doi.org/10.1039/C4CE02324B, https://doi.org/10.1002/adfm.202102868).  In my previous review (comment 1) I suggested to use the term of formation of hierarchical MOF, however the authors did not say anything about it. In general, the explanation of the mesoporosity in the material is weak and need to be clarify.

  1. I still don’t understand the meaning of this sentence in the abstract “dispersed MOFs micropore was expanded into an ordered mesoporous structure by CTAB.”
  2. PXRD revealed that MCS have very low crystallinity and claimed that “all diffraction peaks of MAS, MC, and MCS were in accordance with the standard literature values” is not true at all. For me it’s clear that the MOF is amorphized after the post synthetic modification. Indeed, the pristine MOF prepared in presence of CTAB presented poor crystallinity. As the authors may know, the main diffraction peaks of MIL-101 are in the range of 2-theta 2-4o. Can the authors provide all XRDs including this range? In this way we can see more clearly if the MOF preserve their structure after functionalization.

Author Response

I have carefully read the response letter and the resubmitted manuscript and, I still have some concerns about the work and do not suggest to publish it at the present form.

Response: Thank you very much for your valuable comments and suggestions. All the raised major issues have been clarified and resolved carefully. We believe that the revised manuscript meets the journal's standards and requirements. It would be highly appreciated if you could kindly consider our manuscript for publication.

Indeed, my biggest concern is related with the fact that the authors claimed that they obtained mesoporous MIL-101 when the synthesis was performed in the presence of CTAB. By definition, a mesoporous MOF exhibit open channels or cavities between 2 nm up to 50 nm (at structural level).

As I commented in my previous review, MIL-101 is considered itself as a mesoporous MOF since in its structure exhibits pores and cavities up to 30 A.  By claiming that using CTAB we can obtain mesoporous MOFs could be misunderstanding by the readers. However, by reading and checking carefully the characterization provide by the authors, I strongly believe that interparticle mesopores are formed due to the aggregation of the obtained nanoMOF. This phenomena is not new and it was widely observed when nanoMOFs are synthesized, especially in presence of surfactants (DOI   https://doi.org/10.1039/C4CE02324B, https://doi.org/10.1002/adfm.202102868).  In my previous review (comment 1) I suggested to use the term of formation of hierarchical MOF, however the authors did not say anything about it. In general, the explanation of the mesoporosity in the material is weak and need to be clarify.

Response: Thank you very much for your kind comments. We are sorry for our negligence, and the term of formation of hierarchical MOF has been included in the revised manuscript (including title). As you kindly mentioned, the formed mesopores may be intergranular but not in the real sense, which is formed by the aggregation of nanomaterials obtained. The literature kindly provided by the reviewer confirms this statement (CrystEngComm 2015, 17, 1693-1700; Advanced Functional Materials 2021, 31, 2102868-2102875. Numbered as [36,36]). It has been widely observed when nano-MOFs are synthesized in the presence of surfactants. MIL-101 was shaped into nanomaterials with smaller particles and lower crystallinity under the condition of CTAB, and then the nanomaterials were aggregated to obtain hierarchical porous MOFs. Many thanks.

  1. I still don’t understand the meaning of this sentence in the abstract “dispersed MOFs micropore was expanded into an ordered mesoporous structure by CTAB.”

Response: Thank you very much for your kind comments. We have changed the statement. MIL-101 was shaped into nanomaterials with smaller particles and lower crystallinity under the condition of CTAB, and then the nanomaterials were aggregated to obtain hierarchical porous MOFs (Page 1). Thank you.

  1. PXRD revealed that MCS have very low crystallinity and claimed that “all diffraction peaks of MAS, MC, and MCS were in accordance with the standard literature values” is not true at all. For me it’s clear that the MOF is amorphized after the post synthetic modification. Indeed, the pristine MOF prepared in presence of CTAB presented poor crystallinity. As the authors may know, the main diffraction peaks of MIL-101 are in the range of 2-theta 2-4o. Can the authors provide all XRDs including this range? In this way we can see more clearly if the MOF preserve their structure after functionalization.

Response: Thank you for your comments. It is undeniable that the smaller particle size and lower crystallinity in the sample lead to weaker and wider diffraction peaks. We have adjusted the XRD pattern accordingly. We believe that the position of the main diffraction peak of MCS in the XRD pattern remains unchanged, indicating that MOF retains its structure after functionalization, but its crystallinity is poor (Page 4). Thanks.

Reviewer 4 Report

The revised version is almost ready for publication. However, references 15 and 34 are same. So, Reference 15 may be replaced with "ACS Catalysis, 2019, 9, 1081-1102" before its acceptance.  Also, check the spelling of "particle" in Figure 5 legend.

Author Response

The revised version is almost ready for publication.

Response: Thank you very much for your comments and kind recommendation. All the raised major issues have been clarified and resolved carefully, which have been listed below.

  1. However, references 15 and 34 are same. So, Reference 15 may be replaced with "ACS Catalysis, 2019, 9, 1081-1102" before its acceptance.

Response: Thank you very much for your kind comments. We are very sorry that the two references 15 and 34 are the same due to our negligence. We have replaced reference 15 with the alternative one (ACS Catalysis, 2019, 9, 1081-1102, numbered as [15]) (Page 4). Thanks.

  1. Also, check the spelling of "particle" in Figure 5 legend.

Response: Thank you very much for your kind comments. We are sorry for our negligence of relevant writing issues. We've corrected it to "particle" (Page 6). Thank you.

Round 3

Reviewer 1 Report

The authors have made the necessary adjustments and provided adequate commentary in the updated paper, and most of my concerns have been resolved. The manuscript is now finished and ready for publication.